# Research Progress on Typical Quaternary Ammonium Salt Polymers

**DOI:** 10.3390/molecules27041267

**Published:** 2022-02-14

**Authors:** Xingqin Fu, Yuejun Zhang, Xu Jia, Yongji Wang, Tingting Chen

**Affiliations:** 1School of Chemical Engineering, Nanjing University of Science and Technology, Nanjing 210094, China; fxqnjust@163.com (X.F.); jiaxu@njust.edu.cn (X.J.); wyongji2010@126.com (Y.W.); ctt09011@163.com (T.C.); 2College of Materials and Chemical Engineering, West Anhui University, Lu’an 237012, China

**Keywords:** quaternary ammonium salt, diallyl dimethyl ammonium chloride, acryloyl oxygen ethyl trimethyl ammonium chloride, acrylamide propyl trimethyl ammonium chloride, polymer, intrinsic viscosity, cationicity

## Abstract

Quaternary ammonium salt polymers, a kind of polyelectrolyte with a quaternary ammonium group, are widely used in traditional and emerging industries due to their good water-solubility, adjustable cationicity and molecular weight, high efficiency and nontoxicity. In this paper, firstly, the properties and several synthesis methods of typical quaternary ammonium salt monomers were introduced. Secondly, the research progress on the synthesis of polymers was summarized from the perspective of obtaining products with high molecular weight, narrow molecular weight distribution and high monomer conversion, and special functional polymers. Thirdly, the relationships between the structures and properties of the polymer were analyzed from the perspectives of molecular weight, charge density, structural stability, and microstructural regulation of the polymer chain unit. Fourthly, typical examples of quaternary ammonium salt polymers in the application fields of water treatment, daily chemicals, petroleum exploitation, papermaking, and textile printing and dyeing were listed. Finally, constructive suggestions were put forward on developing quaternary ammonium salt polymers with high molecular weights, strengthening the research on the relationships between the structures and their properties and pinpointing relevant application fields.

## 1. Introduction

Water-soluble polymers are a kind of strong hydrophilic macromolecule that can dissolve or swell in water to form aqueous solutions or dispensing systems. Because of the changeable features of their molecular weight, hydrophilic strength and quantity of feature function groups, water-soluble polymers are widely used in papermaking, water treatment, oil drilling, daily chemicals, and other traditional areas; they also play a key role in strategic emerging industries such as medicine, photovoltaics, artificial intelligence and new energy materials [1,2,3,4,5].

According to the ionic properties of hydrophilic groups, water-soluble polymers can be divided into cationic, anionic, amphoteric and non-ionic types. Cationic polymers herein can usually be formed by homopolymerization, copolymerization with other monomers or modification of other typical function groups based on their monomers [6,7]. Because of their characteristics, including their positive charge, good water solubility, easy control of molecular weight, high efficiency, and low toxicity, water-soluble polymers are widely used in traditional and emerging industries. Cationic monomers are mainly divided into three categories: tertiary sulfur salts [8], quaternary phosphonium salts [9] and quaternary ammonium salts [10]. Quaternary ammonium salts, in particular, are low-cost, have a wide range of raw materials which can be used as their source and are safe. These characteristics have led to quaternary ammonium salts becoming one of the most-studied and widely used product types of cationic monomers.

Among all quaternary ammonium salt monomers, diallyl quaternary ammonium salt, acryloxyalkyl quaternary ammonium salt and acrylamide alkyl quaternary ammonium salt have been the most widely studied. Diallyl quaternary ammonium salt is represented by diallyl dimethyl ammonium chloride (DMDAAC); acryloxyalkyl quaternary ammonium salt is represented by acryloyl oxygen ethyl trimethyl ammonium chloride (DAC) and methyl acryloyl oxygen ethyl trimethyl ammonium chloride (DMC); and acrylamide alkyl quaternary ammonium salt is represented by acrylamide propyl trimethyl ammonium chloride (APTAC) and methyl acrylamide propyl trimethyl ammonium chloride (MAPTAC) in this paper. The classification of cationic monomers and structures of representative quaternary ammonium salt monomers can be seen in Figure 1.

In this paper, firstly, the synthesis methods of the typical quaternary ammonium salt monomers and their polymers are introduced. Then, the relationships between the structures and properties of the polymers are analyzed. Finally, typical examples of the investigation on quaternary ammonium salt polymers in the application fields are listed.

## 2. Typical Quaternary Ammonium Salt Monomers

### 2.1. Properties and Synthetic Methods of the Monomer DMDAAC

DMDAAC is an excellent water-soluble quaternary ammonium salt that contains two unsaturated double bonds; it is a white acicular crystal and has a melting point of 146~147 °C. It is soluble in acetone, methanol, methyl-2-pyrrolidone, tetramethyl urea and dimethyl formamide. At room temperature, its solution is very stable. Furthermore, it is non-hydrolytic, non-flammable, less irritating to the skin and has low toxicity. Since its polymerization reaction activity is low and a small amount of cross-linking reaction always exists, it is difficult to obtain a high molecular weight product [11]. This causes the application fields of its polymers to be mainly focused on raw water treatment, daily chemicals, and oil drilling [12,13], depending on its toxicity and structural stability.

Dimethylamine has been commonly used as a raw material to synthesize DMDAAC, which goes through two reaction steps of tertiary amination and quaternarization of dimethylamine with allyl chloride, as shown in Figure 2. Tertiary amination is associated with many side effects and forms a series of by-products. In industry, the synthesis processes are divided into two-step and one-step methods according to whether the intermediate product, i.e., dimethylallyl tertiary amine, must be separated out for purification and then put through a next step or not. 

The two-step method for intermediate tertiary amine purification after separation and distillation results in a high-purity monomer product. This leads to good polymerization performance. Therefore, the two-step method has become the main method of monomer preparation in the laboratory. In 1951, Butler G B [14] first reported the synthesis of diallyl quaternary ammonium salt and its polymer, in which the synthesis of the quaternary ammonium salt via the two-step method was adopted. After the oil-water separation to obtain the tertiary amine, the quaternary ammonium salt was obtained via its drying, fractionation and reaction with haloalkane in an acetophenone solution.

The synthetic process of the monomer was typically conducted as follows. In 1967, Harada S [15] added an allyl chloride and sodium hydroxide solution into a dimethylamine solution in order and dropwise. Then, the residue allyl chloride and more of the solid sodium hydroxide were added. All procedures were carried out under the condition of stirring at room temperature and reacted for 1 h. The oil phase of the upper layer was separated, dried over solid sodium hydroxide, and distilled to obtain 59~62 °C fractions as a tertiary amine crude product. After drying and fractionation again, 61.5~62.5 fractions were taken as a tertiary amine of the intermediate product, with a yield of about 70%. The tertiary amine and allyl chloride refined by distillation were reacted in distilled acetone for 72 h, and the crystals were filtered to obtain the DMDAAC crystal product, with a yield of about 92.8% after rinsing with cold acetone and drying under vacuum at 60–70 °C. Thus far, this two-step method for the synthesis of DMDAAC has basically continued to be carried out using the above processing route. No obvious substantive breakthroughs have been made since then. The obvious disadvantages of the two-step method include the complexity of the process, the low product yield, and the need for a large amount of organic solvent to be consumed. These disadvantages make the production process costly and unclean; it was not actually industrialized practically until recently [11].

The one-step method is simple, mature, and provides high yield characteristics. It has been widely used in industrial production. However, as an obvious disadvantage, the product obtained by this method contains a large number of by-products. Furthermore, the subsequent purification process is complex, resulting in difficulty obtaining a high purity monomer product, which seriously affects its further polymerization performance [12,16,17].

For example, Hunter [12] prepared DMDAAC by adding all raw materials at one time and reacting these materials under 0.2MPa pressure and a temperature of 54~60 °C. Then, using this monomer as a raw material, the polymer PDMDAAC was prepared with the highest intrinsic viscosity of 0.31 dL/g (test conditions were not specified). It was also considered that the reaction temperature and feeding mode had a distinct influence on the generation of impurities in the monomer product. To reduce impurities and obtain a high-purity monomer, some researchers have adopted the method of lowering the temperature and increasing pressurization. This is intended to prevent the reactants from spillover and reduce the generation of by-products. Even more researchers have adopted the alternating addition of raw materials to prevent the hydrolysis of allyl chloride and other side effects. These researchers, undoubtedly, have made a certain amount of progress, but there are still some other problems, such as an excessively long reaction time at a low temperature, aggravated adverse reactions in a heat-sealed reaction system or a complicated operation procedure using the alternate feeding method [16,17].

In 2019, Zhang Y J [18] purified the monomer DMDAAC product via an improved one-step method and obtained a high-purity monomer DMDAAC product by adding near stoichiometric alkali into the monomer solution to release amines, neutralize tiny excess alkali, and dehydrate and remove salt formed by recrystallization. The content of NaCl was ≤10 mg/kg monomer. Dimethylamine, dimethylamine hydrochloride, dimethylallylamine, dimethylallylamine hydrochloride, allyl chloride, allyl alcohol and allyl aldehyde were not detected. Thus, a new process for high-purity monomer manufacturing by an industrial one-step method was established.

### 2.2. Properties and Synthetic Methods of the Monomer DAC (or DMC)

DAC (or DMC) can dissolve in water, methanol, and ethanol, but not in acetone, ester, or hydrocarbons. Most of its industrial products are aqueous solutions with a weight concentration of about 80%. Because the monomer contains a highly active vinyl group far from a positive charged quaternary ammonium salt, i. e., an electron-withdrawing group, it can be used to synthesize high molecular weight polymers under certain conditions. The products obtained by its homopolymerization or copolymerization have good conductivity and a certain temperature and salt resistance, which allows them to be used as one kind of cationic polymer flocculant widely used in the fields of wastewater and sludge treatment, oil exploitation, papermaking and textile printing and dyeing [3,19,20,21].

DAC (or DMC) is prepared by the quaternization of dimethyl amine ethyl acrylate DMAEA (or methyl dimethyl amine ethyl acrylate DMAEMA) and monochloromethane. DMAEA (or DMAEMA) as an intermediate can be obtained by the alkyl acid esterification method, the methyl acryloyl chloride esterification method, the high-temperature pyrolysis method and the transesterification method. Notably, the transesterification method is the most important method for the preparation of DMAEA (or DMAEMA) and is also the method used in current industrial production. In this method, acrylate or (methyl acrylate) and dimethyl amino ethanol react with each other to form the intermediate via a transesterification reaction under a catalyst, and therefore, the reaction core is the selection of the catalyst [22]. The representative synthesis routine of DAC (or DMC) is shown in Figure 3.

In 2004, Mainz et al. [23] summarized the existing research and found that the disadvantages of this method were the relatively long reaction time (5 h), the appearance of many by-products (91% purity) when using K_2_CO_3_ as a catalyst, the ease of causing a double bond addition as a side-reaction when using an alkali metal catalyst, the sensitivity to water itself and need for the product to be purified by distillation when using a titanium alkoxide catalyst, the low conversion rate (83.5%) when using K_3_PO_4_ as a catalyst, and the long reaction time (8 h) with 95% conversion rate when using dibutyltin oxide as a catalyst.

In addition, other catalysts may be used, including alkali metal oxides and metal alkoxy compounds. When using zirconium acetylacetone as a catalyst, the purity of the tertiary amine ester product was 98.89%, and other impurities such as methyl methacrylate were 0.04%. Under the same conditions, the purity was 93.30% when using dibutyl stannic oxide as a catalyst. The purity was only 86.09% when using tetraisopropyl titanate as a catalyst.

The quaternization reaction usually uses 75% solution of DMAEA (or DMAEMA) and CH_3_Cl under atmospheric or pressurized conditions to prepare an 80% DAC (or DMC) aqueous solution. This method has effectively solved the mass and heat transfer in the reaction process, and the post-treatment of the crude product is relatively simple. The product quality was significantly better than the products obtained by direct reaction of DMAEA (or DMAEMA) with CH_3_Cl or reaction in organic solvents [24,25,26]. Because DAC (or DMC) is highly active and easy to self-polymerize, a certain amount of some polymerization inhibitor, such as phenothiazine, hydroquinone or p-hydroxyanisole, is usually added to prevent self-polymerization.

### 2.3. Properties and Synthetic Methods of the Monomer APTAC (or MAPTAC)

The structure of APTAC (or MAPTAC) is similar to that of DAC (or DMC), but there is a critical difference in that the amide bond in the former has replaced the acryl oxygen bond in the latter, which has overcome an easily hydrolysable disadvantage of DAC (or DMC) monomers by ester groups and led to an even broader application scope. Thus, APTAC (or MAPTAC) has become a cationic monomer with a broad prospect in production for cationic polymer products both of high molecular weight and with valuable performance in heat- and pressure resistance and acid- and alkali resistance.

APTAC (or MAPTAC) is usually obtained through the quaternary ammoniumized reaction of dimethyl amine propyl acrylamide, DMPA (or dimethyl amine propyl methyl acrylamide, DMPMA), and halogenated hydrocarbons. This reaction condition is mild and can be industrially operated with a high monomer yield. For example, Humbert [27] in 1985 dissolved methane chloride and DMPMA in acetone, and they reacted below 50 °C to prepare MAPTAC, with the yield reaching 93.18%. However, DMPA (or DMPMA) was one of the key materials in monomer synthesis.

Generally, DMPA is prepared by the amidation of *N*,*N*-dimethyl propylene diamine and acrylic acid or its derivants (including acrylamide, acryloyl chloride or methyl acrylate) under catalytic action. DMPMA is prepared by the amidation of *N*,*N*-dimethyl propylene diamine and methyl methacrylate under the catalytic action of metal-organic compounds. However, the synthesis of intermediate DMPA (or DMPMA) is difficult due to its harsh reaction conditions and low yield, resulting in high costs for the synthesis process of APTAC (or MAPTAC) [28,29,30]. The representative synthesis routine of APTAC (or MAPTAC) is shown in Figure 4.

In 1984, Makita Muneharu [31] used *N*,*N*-dimethyl propylene diamine and acrylic acid as raw materials, toluene as a solvent, and 4-aminophenol as an inhibitor to prepare DMPA at 180–185 °C for 2 h. DMPA was finally obtained after heating and reflux-dehydrating. The results showed that the industrial yield of this reaction was low, only 54.8%, and there were many side reactions. In 1994, Takao Yuuichi et al. [32] used methyl acrylate and *N*,*N*-dimethyl propylene diamine as raw materials, methanol as a solvent, methyloxy hydroquinone as a polymerization inhibitor, calcium hydroxide as an amination catalyst and concentrated sulfuric acid as a pyrolysis catalyst to obtain DMPA. This method achieved a yield of 77%. It is worthwhile to pay attention to the investigation of synthesis techniques of DMPA, especially those focused on the improvement of yield, the reduction of byproducts and the environmental friendliness of processing.

In contrast to preparing DMPA, the synthesis of DMPMA is relatively not so difficult. In 1979, Franz [33] added *N*,*N*-dimethyl propylene diamine quickly into a mixed boiling solution of methyl methacrylate and catalyst, and then added excessive *N*,*N*-dimethyl propylene diamine at the reflux head temperature of 70~90 °C for 90 min. They then gradually heated the solution to 150 °C. DMPMA was obtained with a 92% yield by removing unreacted methyl methacrylate in a vacuum at 130 °C.

In 2014, Liu Z Q [33] used methyl methacrylate and *N*,*N*-dimethyl propylene diamine to synthesize DMPMA in inert organic solvents such as toluene. Linear fat hydrocarbons were used to remove the methanol by using the zeotropic nature of the organic solvent and methanol. With this method, the yield of DMPMA could reach more than 95%. However, due to the use of a large amount of poisonous organic solvent, there were obvious shortcomings in environmental protection and safety.

## 3. Research Progress on the Synthesis of Quaternary Ammonium Salt Polymers

Quaternary ammonium salt polymers, according to their polymerization types, mainly include the homopolymer of the cationic monomer, the binary copolymer of the cationic monomer and other monomers like acrylamide (AM), the multiple copolymer of more than two monomers, and modified polymers. According to the polymer structure, they can be divided into linear polymers, micro-crosslinked polymers and crosslinked polymers. For linear polymers, which are one of the most widely synthesized and applied kinds, the molecular weight of the polymer products is used as an indicator in evaluation. Sometimes, the molecular weight distribution, monomer conversion and sequence structures are also used for specific purposes. Crosslinked polymers, however, are usually polymerized under the guidance of special functions for some application. This section summarizes the research progress on the synthesis of homopolymers and binary copolymers with high and serially intrinsic viscosity (characterization of molecular weight). Meanwhile, a brief overview of the research on polymers with narrow distributions of molecular weight, high monomer conversion rates in the product, and special function groups in the polymer chain is outlined. Here, the schematic diagram of cationic monomers, their polymerization processes and polymer products is shown in Figure 5.

### 3.1. Synthesis of Polymers with High and Serial Intrinsic Viscosity

A quaternary ammonium salt polymer usually refers to a type of typical linear polymer. A linear polymer has different chain lengths characterized by different molecular weights or intrinsic viscosities which correspond to different properties and application performances. The improvement of or increase in the intrinsic viscosity means the possibility of synthesizing polymers with broader scopes and more grades in their serial intrinsic viscosity, which means that better polymer products with wider application performances can be achieved. Therefore, from the perspective of synthesis and processing research, how to obtain polymers with high intrinsic viscosity has been a very important purpose for the long term, and many researchers have been engaged continuously in this with long-term and in-depth work.

#### 3.1.1. Progress in the Synthesis of DMDAAC Polymers

For the ring closure polymerization of DMDAAC, its activity is impaired due to both the strongly electron-withdrawing effect of quaternary ammonium salt on the electronic cloud in double bonds with a short distance to the salt and the steric hindrance needed to be overcome in the ring closure process. Meanwhile, at an early time, limited by the level of synthesis technique, there were many by-products as impurities in the monomer product, leading to difficulty for researchers to obtain homopolymerization and copolymerization products with higher molecular weight [34,35].

Since 1956, studies on the synthesis of homopolymer PDMDAAC have been reported. The initial target polymer was a three-dimensional network product, but the obtained polymer was, in fact, a linear polymer of extremely low molecular weight with ring structure units as its chain. The typical representative of a homopolymer successfully synthesized in the early stage was made by Boothe J E. [34] In 1970, the various influences of polymerization conditions and impurities on the intrinsic viscosity of the product, PDMDAAC, were investigated by using APS as the initiator. The maximum intrinsic viscosity value was 1.80 dL/g (determined in 1.0 mol/L NaCl solution and calculated, the same as below without explanation). Since then, more and more researchers have carried out in-depth studies on it. In 1999, Ransohof J A [36] took a commercially available monomer as a material, 60 mL syringe as the reactor and a concentration of 72.0% monomer. The product, with an intrinsic viscosity of 7.7 dL/g PDMDAAC (measured in 2.0 mol /L NaCl solution and calculated), was prepared after 20 h reaction initiated by 35 kilorads/h Co-60 ray. This is currently the maximum value of PDMDAAC regarding intrinsic viscosity that has been obtained in literature reports. However, due to the small reactor and the radiation polymerization method, no large-scale industrial production has been reported so far. In 2016, Zhang Y J [37] synthesized homopolymer PDMDAAC by using a high-purity industrial monomer and the three-step temperature raising process, and the intrinsic viscosity values of the products reached 3.99 (short time) and 4.70 dL/g (prolonged time), respectively, which are currently the highest values for products obtained by industrializable processing methods.

The typical and widely used copolymer of DMDAAC is the copolymer product of DMDAAC with AM, i.e., P(DMDAAC-AM), referred to as PDA. In 1960, its synthetic method was first reported [38]. Then, from the mid-1980s to the early 1990s, the Research Institutes of Chemical Industry in Beijing and Zhejiang in China began to investigate the PDA copolymer. In the last twenty years, there has been increasing research on the structure, processing, and application of PDA [35,39,40]. Specifically, when high-purity industrial product DMDAAC and AM are used as raw materials and a metal chelating agent and initiator were added to initiate polymerization, which was then maintained for 9 h, the colloidal products were generated with 5–50% cationicity. Furthermore, the intrinsic viscosity was 21.0–7.0 dl/g, which is currently the serial and highest intrinsic viscosity of PDA reported in the literature [41].

#### 3.1.2. Progress in the Synthesis of DAC (or DMC) Polymers

Compared with the molecular structure of monomer DMDAAC, the positive charge center in the structure of monomer DAC (or DMC) is far away from the double bond, which makes it easy to form polymers with high molecular weight via polymerization. Furthermore, by comparing the two monomers, it can be seen that DMC has one more methyl on the double bond than DAC, making it more stable. This also causes the reporting time of DMC to be earlier than DAC.

In 1976, Nishikaji T [42] first synthesized monomer DMC, which then led to a boom in the synthesis and application research of its polymer, and soon established a pattern dominated by copolymer products of DMC [43,44,45]. After the first synthesis of PDMC with K_2_S_2_O_8_ as an initiator by Ringsdorf [46] in 1977, the research of PDMC was focused on linear polymer for 30 years. However, the high molecular weight of the polymer product was limited to 9.5 dL/g [47]. In 2015, Tang Li et al. [48] in our group used a two-step heating method by using KPS as an initiating system and synthesized PDMC with an intrinsic viscosity of 14.7 dL/g. At this point, the synthesis technique of polymer PDMC had made a new breakthrough, the molecular weight of PDMC had been greatly improved, and the synthesis process was easy to be industrialized.

At present, the typical copolymer of DMC is still the copolymer P(DMC-AM). There are numerous studies about the synthesis of P(DMC-AM), but in the first 20 years after its successful synthesis, the research focused on the low cationicity and the product intrinsic viscosity was generally limited. This led to the understanding that P(DMC-AM) cannot meet with requirements of high-sheared flocculation for some dewatering occasions. Therefore, gradually, another kind of cationic polymer P(DAC-AM) replaced its market dominance. Over more than 20 years, however, many researchers investigated how to obtain a high intrinsic viscosity polymer by adjusting the process parameters in the polymerization reaction system [49,50,51]. In 2019, based on the processing characters used by Tang Li in the homopolymerization process [48], our group established a new synthetic method for P(DMC-AM) to obtain products with serialized cationicities of 10~90% and corresponding high molecular weights of 18~14 dL/g [19], which were the highest intrinsic viscosity values reported for these products with the same cationicity.

With the development of science and technology and the increased importance of environmental protection, the molecular weight of cationic polymers used as a flocculating and dehydrating agent has been gradually required to be higher. Interestingly, when comparing DAC with DMC, the double bond in DAC had one less methyl group than DMC, making the double bonds in the polymerization a much lower steric hindrance. Therefore, higher molecular weight polymers based on DAC could be synthesized. Its requirements in markets at the end of the 20th century exceeded the polymer of DMC and dominated a leading position in the world market [50,52,53]. Our group successfully synthesized homopolymer PDAC using industrial DAC as the raw material by means of a method named temperature program-increasing for initiation and -maintaining for polymerization. The intrinsic viscosity of the PDAC obtained was 14.61 dL/g [54], which is currently the highest value reported in the literature.

For the copolymer of DAC and acrylamide AM, Cabestany et al. [55] first synthesized P(DAC-AM) on the laboratory scale, which was soon put into industrial production and application. In 2019, Zhang et al. [56] used industrial monomer DAC and AM as raw materials and redox initiator to synthesize P(DAC-AM). This P(DAC-AM) demonstrated 10~90% cationicities and corresponding intrinsic viscosities of 26~15 dL/g, which are currently the maximum values reported.

#### 3.1.3. Progress in the Synthesis of APTAC (or MAPTAC) Polymers

In the molecular structure of APTAC (or MAPTAC), the positive charge center is far from the double bond, so it also has high polymerization activity. At the same time, the amide group structure can overcome the shortcoming of the ester group in DAC (or DMC) easily to hydrolyze, which makes their polymers have a broader application field and prospect.

Monomer MAPTAC was first successfully prepared by Franz et al. in 1979 [33], but it was not until 2003 that its polymer PMAPTAC with 1.99 × 10^5^
*M_w_* was synthesized for the first time [57]. Then, PMAPTAC with a serialized intrinsic viscosity of 2.0–8.2 dL/g was obtained through the temperature program-increased and -maintaining method by using industrial monomer MAPTAC as a raw material [58]. The obtained intrinsic viscosity of 8.2 dL/g is currently the highest level reported in the literature.

However, the copolymer of MAPTAC was studied earlier. In 1982, Phillips et al. [59] first used AM and MAPTAC as raw materials and AIBN as an initiator to obtain P(MAPTAC-AM) with 10% cationicity and an intrinsic viscosity of 11.4 dL/g by a two-step heating process. There were many literature reports on its copolymerization with acrylamide after that [60,61]; however, the cationicities of the products were low and single, and the corresponding intrinsic viscosities were also low. In 2014, Li et al. [62] prepared polymers with 5–50% cationicities and intrinsic viscosities of 17.2–9.7 dL/g by using AM and MAPTAC as raw materials through a three-step heating process. This paper was the first report on the copolymer P(MAPTAC-AM) with a series of cationicities and intrinsic viscosities. Then, Bi et al. [63] further expanded the range of cationicities of the copolymer, obtaining copolymers with 10~90% cationicities and corresponding intrinsic viscosities of 13.0–2.58 dL/g.

In 1998, Mallon first synthesized APTAC, and then researchers conducted in-depth studies on its homopolymer and copolymer. In 2014, Li et al. [62] synthesized copolymer P(APTAC-AM) with 13~75% cationicities and corresponding intrinsic viscosities of 21.5–9.7 dL/g by adding all raw materials of APTAC, APS initiator and Na4EDTA at one time with a three-step temperature program-increasing method. Later, in 2019, PAPTAC, the homopolymer with high intrinsic viscosity, was prepared by a similar method [64].

#### 3.1.4. Synthetic Processes and Products of Quaternary Ammonium Salt Polymers

In summary, for a cationic and linear polymer, the cationic unit structure is the primary element that dominates the essential properties of the polymer. Once the structures of monomers are determined, their proportions in the polymer (the cationicity) become the main factor that dominates the basic properties of the polymer. When the above two core elements were determined, the acquisition of polymers with a high molecular weight through the control and optimization of polymerization conditions became a common goal that researchers pursued for a long time. Synthetic processing of an obtained high molecular weight polymer could provide raw materials for exploring new properties and functions of the polymer and lay a foundation for obtaining products as raw materials with a wider serialization of molecular weight usable for the in-depth development of application fields. Therefore, over the years, researchers have pursued the improvement of the purity of the monomer to decrease side reactions and facilitate the optimization of the polymerization process, i.e., the appropriate conditions of monomer concentrations and initiator agent systems [59,65,66]. It was, therefore, a huge success for the temperature program-increasing and -maintaining method to prepare the high molecular weights of both cationic homopolymers with a high and widely serial molecular weight and the copolymers with a series of cationicities and high molecular weights. The highest intrinsic viscosities of the representative quaternary ammonium salt polymers are summarized in Table 1.

### 3.2. Studies on Polymers with Narrow Molecular Weight Distribution and High Monomer Conversion

In addition to the ionicity and molecular weight, the importance of the molecular weight distribution and monomer conversion in the polymers cannot be ignored. Molecular weight distribution is a key factor influencing the performance of polymers. Polymers with different molecular weights have different properties, resulting in various performances in different applications. Generally, the polymer with a narrow distribution in a proper molecular weight can better improve utilization and performance to reduce the dose and achieve an efficient economy. Improving the monomer conversion can increase the utilization of a polymer and reduce the possible pollution of a toxic monomer residue while reducing the post-treatment process in applications. Therefore, in a synthetic process, it is of great significance to control or adjust the molecular weight distribution and to improve the monomer conversion of polymers.

#### 3.2.1. Synthesis of Polymers with Narrow Molecular Weight Distributions

In the practical production and actual synthetic process for large-scale productions, the molecular weight distribution is generally not considered a main index in the evaluation but merely a parameter in the characterization of the target polymer, occasionally. However, for some polymers of low molecular weights to have fine functions in application, the determination of a molecular weight distribution coefficient is usually necessary. On the one hand, the reason for this is that polymers with a narrow molecular weight distribution obviously demonstrate some special functions when within a certain low molecular weight. On the other hand, it is difficult for an available instrument, like GPC, to test the molecular weight and distribution coefficient for polymers of a huge molecular weight.

Generally, in traditional polymerization methods, compared with aqueous polymerization, the molecular weight distribution obtained by emulsion polymerization is narrower. For example, in 2020, Yang et al. [19] synthesized P(DAC-AM) by an emulsion polymerization method and controlled the temperature by adjusting the dose of the molecular weight regulator so as to control the polymerization rate. They finally obtained a polymer with a narrow-distributed molecular weight of 1.12 *M_w_*/*M_n_* and 0.4 million. In 2013, Xiong Q Y [67] synthesized 70% cationicity PDA by the aqueous solution polymerization method by controlling the reaction temperature, monomer concentration, initiator concentration and other processing factors. The molecular weight distribution coefficient was reduced from 2.715 to 1.687.

In addition, RAFT (reversible addition-fragmentation chain transfer) polymerization, which is characterized by a series of addition-fracture equilibria, was proposed to obtain polymers with a narrow distribution. The key factor in this process is the added chain transfer agents [68,69,70]. During the polymerization process, the functional groups of the chain transfer agent facilitate free radical exchange and act as a buffer to keep the concentration of active free radicals at a low level and to control the growth of the active chain in the process of free radical polymerization. Finally, a narrow distribution polymer could be obtained. In 2016, Ohno et al. [71] used the RAFT method to synthesize PMAPTAC with *M_n_* and *M_w_*/*M* of 2.61 × 10^4^ and 1.10, respectively. This was accomplished by adding the chain transfer agent and initiator dissolved in methanol into the MAPTAC solution for polymerization. Huang et al. [72] synthesized P(DMC-AM) with a distribution coefficient of 1.1~1.2 by RAFT polymerization using 3-benzyltrithiocarbonyl propionic acid as a chain transfer agent. There are many similar reports [73,74,75] that are characterized by mild reaction processes, the narrow molecular weight distribution of products, and the low molecular weight of the synthesized polymers.

#### 3.2.2. Synthesis of Polymers with High Monomer Conversion

In practical synthetic processes, few studies used the monomer conversion as the main evaluation index; most studies used monomer conversion only as an auxiliary index. Generally, adjusting the reaction condition parameters during the synthesis process of polymers can not only reduce the molecular weight distribution coefficient, but also improve the monomer conversion. For example, in 2016, when synthesizing PDMDAAC, Zhang et al. [66] found that monomer conversion could be increased by selecting the polymerization temperature and prolonging the polymerization time. In 2021, Zhang et al. [76] adjusted monomer concentrations to 30.0–60.0% and azo initiator concentrations ranging from 0.85% to 0.06%. PDA with serial molecular weights of 2.00~14.00 dL/g were obtained. Their monomer conversions were all above 99.50%.

### 3.3. Synthesis of Polymers with Special Functions

With the rapid development of science and technology, the application field of cationic polymers such as the quaternary ammonium salts has been continuously expanded. When designing and synthesizing the target polymers, apart from the traditional indexes such as the molecular weight, the special functions of polymers have been increasingly usedas the guidance for polymerization.

To obtain polymers with multiple properties and performances, many researchers synthesized multipolymers by using a variety of monomers. For example, in 2018, Luo et al. [77] synthesized a multipolymer by using methacrylic acid, sodium methylallyl sulfonate and DMDAAC as materials via a free radical copolymerization in order to improve the absorption rate of chromium-free leather for anionic dyes, retaining agents and fat liquors. The results showed that the multipolymer had strong adsorption ability, and the adsorption rate of dyes and fat liquors was over 96%.

In recent years, studies on the molecular brushes [78,79] and material carriers of polymer [80,81] have been carried out by using cationic monomers and through a crosslinking reaction, attempting to use them in the fields of life science and biosensors. For example, in 2018, Kou et al. [78] synthesized a PDMC molecular brush with pH response by RAFT polymerization on a gold coating layer pre-treated with lotion and studied the changes of hydration, conformation and surface wettability of the PDMC molecular brush under different pH values.

The investigation of modification of, or grafting on, a polymer chain to explore its new uses or efficiency has been a research hotspot for many years [82,83,84,85]. For example, Wang et al. [86] synthesized a lignin-DMC-grafted polymer under a short-wavelength UV light and studied various technological conditions (such as the mass ratio of DMC/lignin to the amount of photoinitiator). The synthesized mixture was precipitated in ethanol solution to separate the lignin-DMC copolymer. Then, the precipitate was washed several times with ethanol and acetone mixture, and finally, the cationic lignin graft polymer was obtained by freeze-drying.

## 4. Studies on the Relationships between Structures and Properties of Polymers

The properties of a compound are usually determined by its components and structure. In terms of linear quaternary ammonium salt polymers, if the structure of the positive charge unit and its proportion are certain, its characteristic properties and main functions are basically determined, such as its electric neutralization ability and structural stability. Furthermore, when controlling their molecular weights to be different, many properties and performances of the polymers also vary, such as their dispersion and flocculation functions. In addition to the charge density and molecular chain length, in some applications, changes in the molecular weight distribution and the microstructure of the cationic unit of the polymers may also result in significant functional differences.

### 4.1. Molecular Weights and Application Performance

Linear cationic polymers have different chain structures and lengths which correspond to different structural properties and application functions. When the structure and proportion of the chain units are constant, i.e., the polymer type and charge density are the same, the properties and functions depend on the molecular weight.

Homopolymers of quaternary ammonium salts generally have relatively lower molecular weight and are often used as dispersants, reinforcements, fungi- or bactericides in papermaking, textile printing and dyeing, and daily chemicals in other industrial fields, including raw water treatment [87,88,89]. In 2010, Yu et al. [87] synthesized PDMDAAC with a molecular weight of 0.18–0.81 dL/g and measured its color fixing performance. It was found that when its molecular weight was 0.24–0.47 dL/g, its color fixing performance was better. In addition, the dyeing fastness, soap washing fastness and dyeing effects of the fabric could also be significantly improved by controlling the molecular weight of PDMDAAC. In 2013, Zhao et al. [90] used the composites of 0.55–2.47dL/g PDMDAAC and PAC as the composite coagulants and tested the efficiency of coagulants in killing bacteria in raw water and in sediment sludge through jar tests. The results showed that under the same dosage, with increased molecular weight under the same dosage, the bactericidal rate increased from 88.32% to 91.11%, indicating that the increased molecular weight could improve the bactericidal performance.

After a copolymerization of a cationic monomer and AM, the significant characteristic is that the molecular weight becomes greatly increased. This can often be used as a flocculant dehydrating agent for the treatment of wastewater and sludge dehydration and as a drilling fluid and fracturing fluid for petroleum exploitation [5,19]. For example, in 2018, Djibrine et al. [50] synthesized P(DMC-AM) polymers with molecular weights of 5.0 × 10^6^ and 7.0 × 10^6^ and used them to conduct flocculation experiments on kaolin wastewater. The results showed that the residual turbidity and the particle size d_50_ of the lower molecular weight polymer were 10.1 NTU and 468 μm, respectively, while the higher molecular weight polymer was 5.9 NTU and 566 μm, respectively.

Therefore, it can be seen that the molecular weight required in different application fields is not the same. The synthesis of polymers with a serial molecular weight could not only give full play to its efficiency but also reduce the use of dosage to save raw materials, i.e., reduce waste discharge and protect the environment.

### 4.2. Relationships between Charge Densities and Properties

Cationicity refers to the ratio of the number of cationic units to the total number of units in a polymer. It is also the measurement of molecular charge density. When the cationic unit is determined, the charge density of the homopolymer is done at the same time. Thus, when concerning the charge density, the material should be a copolymer. When the molecular weights of copolymers are closed, charge density must play a larger influence on its performance.

On the one hand, the cationicity directly affects the morphology of a polymer in aqueous solution, because there is a certain repulsion between neighboring cationic charges on the molecular chain, which results in the extension of the molecular chain in aqueous solution [65]. Chen et al. [91] studied the apparent viscosities of P(DAC-AM) with 10%, 50% and 90% cationicity in different salt solutions. The results showed that in the same salt solution concentration, such as NaCl, when the cationicity of the polymer with the same intrinsic viscosity increased from 10% to 90%, the apparent viscosities at the stabilization after the first reductions increased from 7.86 cP to 8.16 cP. The reason, the researchers suggested, was that the greater the cationicity, the greater its repulsion was in the same salt solution; thus, the greater its elongation was, and the higher its apparent viscosity was. On the other hand, the cationicity is directly related to the electrical neutralization capability, conductivity, and bactericidal properties of the copolymer. Generally, the higher the cationicity, the stronger these properties are [92]. However, in practical applications, such as wastewater treatment, it was found that when the cationicity of a given polymer matched the Zeta potential of the actual treated object, the application effect was best, too large, or too small, and not conducive to its flocculation function.

Yang et al. [93] synthesized P(DAC-AM) with cationicities of 10~40% and applied it to treat papermaking wastewater. The experimental results showed that when the cationicity of P(DAC-AM) increased from 10% to 25%, the COD removal rate increased rapidly from 86% to 94%, and the SS removal rate increased from 83% to 92%; when the cationicity increased continuously to 40%, the COD and SS removal rates only increased by 1%. When the dosage increased synchronously above 26 mg/L, contrarily, the COD and SS removal rates decreased slightly, indicating that 30% cationic P(DAC-AM) had the best flocculation effect on dehydration of papermaking wastewater. Chen et al. [94] used P(DAC-AM) with 10~90% cationicities to treat the sludge in a production process of poly-aluminum chloride by an aluminum ash method. The results showed that with the increase of cationicities from 10% to 90%, the solid content of filter cake increased firstly from 52.17% to 62.46% and then decreased to 59.97%. The filtrate volume was increased from 23 mL to 34.5 mL and then reduced to 29.5 mL, and the optimal cationicity of P(DAC-AM) for the application was 70%.

Therefore, it can be seen that different application fields require different copolymer cationicities. The synthesis of a serial cationicity polymer could meet different application requirements and give full play to the product efficiency.

### 4.3. Solution Properties of Polymers

Quaternary ammonium salt polymers, as the most typical class of cationic polymers, are usually applied in the form of a solution, and their performance is always closely related to their properties in the solution state. The solution properties mainly manifest in the viscosity behavior, molecular morphology, and size of polymers, which depend on not only the salt solutions, but also the polymer’s molecular components and structures.

Fu et al. [65,91] investigated the viscosity behavior of P(DMC-AM) and P(DAC-AM) in different salt solutions (LiCl, NaCl, KCl, MgCl_2_, AlCl_3_, Na_2_SO_4_ and Na_3_PO_4_). The results showed that the apparent viscosity in the inorganic salt solutions first decreased sharply to its lowest value with the increase in the salt concentration, remained stable when the added salt was monovalent, and increased with increased salt concentration when the added salts were multivalent. The aggregation model of the cationic polymer in salt solutions after the inflection point is shown in Figure 6 [91].

A given polymer’s molecular morphology is obviously affected by the ionic properties and temperature of its solvents. Under different solvents and temperatures, the extension state of the molecule is different, and the *K* and *α* values of the two coefficients in Mark-Houwink ([*η*] = *KM^α^*, characterizing the relationship between the intrinsic viscosity and molecular weight) are also different. There have been many studies conducted regarding the *K* and *α* values of PDMDAAC [95,96], but none for the other five kinds of typical quaternary ammonium salt polymers. These other typical quaternary ammonium salt polymers frequently use the Mark-Houwink values from anionic polymers [13]. Otherwise, the *K* and *α* values of partially hydrolyzed polyacrylamide have also been studied [97]. They are summarized in Table 2.

### 4.4. Stability of Polymers

Typical quaternary ammonium salt polymers are usually used in aqueous solutions for water treatment, papermaking, textile dyeing and finishing, and other purposes [87,88,89,95]. Sometimes, they are applied in certain high-temperature environments in solid or colloidal form, as is the case for photosensitive materials, temperature-sensitive elements and antistatic coatings on some electronics [5]. The stability of polymers after heating and hydrolysing in aqueous solutions depends mainly on the properties of the hanging structures in their chain units and rarely on the effects of the length of the chain, i.e., the molecular weight.

#### 4.4.1. Thermal Stability

Studies on the stability of quaternary ammonium salt polymers found that the stability of the DMDAAC polymer and APTAC (or MAPTAC) polymer was significantly better than that of the DAC (or DMC) polymer. For example, Jia et al. [95] studied the thermal stability of the PDMDAAC homopolymer and found that the polymer began to perform side chain decomposition at around 288 °C. Su G [98] found that PDMC began to decompose its side chain at about 240 °C. Wei et al. [82], characterizing grafting PMAPTAC, found that the polymer decomposition in the first stage was in the range of 280~340 °C. Thus, it could be seen that DAC (or DMC) polymers started to decompose at a lower temperature than DMDAAC polymers and APTAC (or MAPTAC) polymers; that is, their thermal stability demonstrated a slight weakness.

#### 4.4.2. Hydrolytic Stability

At present, no literature has systematically analyzed and compared the hydrolysis stability of these typical quaternary ammonium salt polymers, including the polymers with serial molecular weight or cationicities. The existing reports were only studies on a single polymer. Lin [99] studied the hydrolysis process of PDA and found that the hydrolysis velocity of the amide group on the AM unit was twice that of the quaternary ammonium salt group on the DMDAAC unit. Under alkaline conditions (10%NaOH), the intrinsic viscosity representing the velocity changed little within 90min, but decreased by 3/4 under the same conditions at 175 °C. In 2019, Wu et al. [100] conducted hydrolysis resistance experiments with the synthetic homopolymers PMAPTAC and PDMC. The polymers were placed in a closed space, the reaction temperature was maintained at 50 °C, and the experiment time was 12–92 h. The results showed that at 65 h, the ester group was hydrolyzed completely, and a carboxyl group was formed. However, PMAPTAC did not hydrolyze in the experimental range at all.

In summary, it can be seen that the hydrolysis degree of quaternary ammonium salt polymers increases with increased solution pH and temperature as well as time. In addition, the DMDAAC polymer and APTAC (or MAPTAC) polymer had significantly better hydrolysis stability than the DAC (or DMC) polymer, and the homopolymer was usually better than the copolymer, which might be caused by the stability of characteristic groups on the units.

### 4.5. Differences in the Microstructures of Polymers

The microstructure differences of a polymer, including differences in the unit structures, chain arrangement and molecular modification, influence the performance of the polymer.

#### 4.5.1. Microstructural Differences of Units

In 2013, Yu et al. [101] designed the modified monomer 3-chloro-2-hydroxypropyl methyl diallyl ammonium chloride (CMDA) by modifying the monomer DMDAAC by replacing a methyl group on the quaternary ammonium salt group with chlorpropyl. The fixatives P(CMDA-DMDAAC) were obtained by the copolymerization of the modified monomer with the reference monomer, DMDAAC, while controlling the unit structure ratio and molecular weight. The results showed that the fixatives with an intrinsic viscosity of 0.26–0.76 dL/g and a CMDA unit content of 8–20% showed more outstanding fixation efficiency and had a wider range of stable performance. Its color fixing performance, especially dry and wet friction fastness, was better by 1 and 0.5 grades, respectively, than commercially available PDMDAAC color-fixing agents, whether domestically made or imported. Therefore, it was a good prospect for application. Here, the unique mechanism explaining these results was the added hydroxyl group, which formed an ether bond with the hydroxyl group on fibers.

In 2019, Zhang et al. [102] modified the monomer DMDAAC. Under the precondition of maintaining the diallylmethyl groups, only one of the original methyl groups on the quaternary ammonium nitrogen was modified with several lipophilic alkyl groups, respectively. Serial homopolymers of methyl alkyl diallyl ammonium chloride, such as propyl, amyl and heptyl, were prepared. The results showed that with the increase in the alkyl group length, the CMC (critical micelle concentration) values were 0.042, 0.030 and 0.020 mol/L, respectively; the corresponding γ_CMC_ values (surface tension at CMC) were 46.84, 41.46 and 31.92 mN/m, respectively; and HLB values (hydro- and lipophilic balance) were 11.65, 10.70 and 9.75, respectively. These results indicated their ability to reduce surface tension and surfactivity, their efficiency to reduce surface tension, and their capacity for their solubility in oil to be increased.

From the perspective of microstructures, DMC and DAC monomers differ only by one methyl in the double bond on the cation unit, so there is little difference in almost all application fields of their corresponding polymers. However, in the experimental study, it was found that the treatment effects of P(DAC-AM) and P(DMC-AM) on wastewater (sludge) with different properties were significantly different. Chen et al. [92,94] conducted flocculation and dehydration experiments by using P(DAC-AM) and P(DMC-AM) as flocculants to treat the waste sludge produced by the polyaluminium chloride production process and the residual activated sludge produced by the cassava alcohol production process after biochemical degradation, respectively. The results showed that when they were used in the treatment of the waste sludge from polyaluminium chloride production, P(DAC-AM) and P(DMC-AM) could both disperse easily in sludge and isolate the water out via destroying the balance of the original aqueous colloid in the sludge to achieve a certain effect of flocculation and dehydration. The values of the cationicity and molecular weight for both polymers used in their best treatment effects were similar. However, the effectiveness of P(DAC-AM) was even more obvious for the sludge cake to be porous and blocky, since the volume of wet residue cakes by P(DMC-AM) treatment and after filtration was reduced by 13%, and 16% more polyaluminum chloride solution products as a filtrate from wet residue cakes were recovered. The outer surface and transverse section of dried sludge filter cake treated by P(DAC-AM) are displayed in Figure 7.

However, for the activated sludge from the cassava alcohol production process, it was found that although large flocs could be formed by using both polymers under the same conditions, the sludge cake after P(DAC-AM) treatment and then filtration was jelly-like and difficult to deeply dehydrate. However, after P(DMC-AM) treatment, flocs could be rapidly and deeply dehydrated to cakes, achieving a good dehydration effect. The simple explanation was that monomer DMC had one methyl more than DAC in its microstructure, which made the DMC polymer a little more lipophilic and provided a better dehydration effect on organic wastewater. The possible flocculation mechanism is displayed in Figure 8. 

#### 4.5.2. Differences of Linear and Branched Structures

Guo et al. [103] synthesized linear and branched DMC polymers and applied them in the flocculation and dehydration of papermaking sludge. The results showed that the decolorization effect of a branched polymer was better than that of a linear polymer, and the maximum decolorization rate could reach more than 95.0%. The flocs formed by branched flocculants were large, loose, and easy to precipitate, while linear flocculants were not. The authors speculated that the flocculants with abundant side chains had a stronger adsorption-bridging effect.

#### 4.5.3. Molecular Structure Modification Based on Natural Polymers

The research on the modification of natural polymers has a long history. Modification can change both the featured and microstructures of a polymer. The modified polymer has the various advantages of biodegradability and a wide range of raw material sources, which can expand the application fields of the original polymer or improve its application efficiency. The modification of a natural polymer matrix by cationic monomers has been a hot research topic for many years.

In 2013, Pourjavadi et al. [104] dissolved starch in water at a stir speed of 200× *g* and heated the solution formed at 80 °C for 30 min under nitrogen atmosphere. They then added MAPTAC and AM as modifying raw materials, added ammonium persulfate as an initiator in the solution after mixing evenly, and continued the reaction for 15 min. After finishing the reaction, the product was dehydrated with ethanol and dried at 50 °C for 24 h in a convection oven. The starch-grafted polymer was synthesized and found to be very effective in settling cement particles, which were successfully used to produce green fiber-cement composites. In 2020, Wang et al. [85] synthesized a partially de-branched starch-DAC polymer with corn starch as a raw material in the presence of hydrogen peroxide and acetylacetone. It was found that the adhesion to polyester/cotton yarn was significantly enhanced. There were many similar literature reports of the graft modification on natural polymers by cationic quaternary ammonium salt monomers [82,83,84,85].

In summary, it can be seen that the application efficiency of polymers can be improved and the application field can be expanded by modifying the microstructure, especially the cationic structure of polymers. Therefore, the precise regulation of polymeric microstructures has always been one direction of related research.

## 5. Research Progress on Applications of Quaternary Ammonium Salt Polymers

As a kind of polyelectrolyte, the quaternary ammonium salt polymers can not only be used in traditional fields such as water treatment, daily chemicals, petroleum exploitation, papermaking, textile printing and dyeing, but also in strategic emerging industries such as pharmaceuticals, photovoltaics, artificial intelligence and new energy materials. Their unit structures, the proportion of cationic units and the molecular weights of polymers are their internal key factors that can be exploited to attain various advantages.

### 5.1. Water Treatment

Because of their adjustable charge density and molecular weight, quaternary ammonium salt polymers are mostly used as coagulants and flocculants in various raw water, wastewater and sludge treatments. When used, they can fulfill multiple functions, such as performing electric neutralization, acting as a compression double electric layer, and facilitating bridging and net capture. At this point, the characteristics of a quaternary ammonium salt monomer and its formed unit structure play a decisive role in its function utilization.

#### 5.1.1. Raw Water Treatment

Tiny-polluted raw water refers to surface water that is polluted by organic matter yet nevertheless exceeds the sanitary standard of some water quality indexes; it contains many kinds of organic matter of complex nature, but their concentrations are not high. With the development of society and the economy and the increase in population density, surface water, especially that used as a drinking water source, is becoming increasingly polluted, even as the people’s demand for drinking water quality is increasing. Therefore, many researchers began to explore new treatment techniques of tiny-polluted raw water in the late 20th century. PDMDAAC, with its high charge density, good stability, and low toxicity, was the first organic polymer approved by the FDA in the USA for use in drinking water production [105].

As early as 1986, Tanaka [106] used a PDMDAAC homopolymer to remove suspended particles in low turbidity water and studied the kinetics of the removal of suspended particles removal by changing the polymer type (PDMDAAC and polyvinyl imine), molecular weight and dosage. The results showed that the medium molecular weight (40 thousand) PDMDAAC had a significant adsorption effect on particles. Later, in 1989, Haarhoff et al. [107] found that PDMDAAC had a good removal effect on both extracellular organic matter and algae cells in the treatment process of algae-containing water.

However, many studies found that, when using PDMDAAC alone for coagulation treatment of raw water, it was difficult for its treatment effect to meet the required turbidity removal requirements, and its use cost was high. Therefore, in the conventional water supply process, it was not used alone but as a coagulant aid with inorganic flocculants. A typical representative work was what Li Xiao Xiao et al. [108,109,110] made. In 2011, a series of stable inorganic salt coagulants/PDMDAAC composite coagulants were prepared and used to enhance coagulation and turbidity-removal treatment aimed at several typical tiny-polluted surface water sources in various locations in China, such as the Yangtze river, Taihu river and inland rivers with different seasons, as well as several typical kinds of raw water that are difficult to treat, such as raw water with low turbidity at a low temperature and organic-polluted and high algae-containing water. The results showed that the efficiency of electric neutralization and bridging represented by turbidity removal and floc sizes increased when using the composite coagulants compared with PAC alone. The enhanced coagulation effect of treatment processes using composite coagulants becomes more and more significant as the degree to which waters are tiny-polluted increases. The increase in the PDMDAAC ratio and intrinsic viscosity, that is, the improvement of the electric neutralization and bridging capacity, also indicated the advantages of composite coagulants in coagulation [108,109,110].

In 2019, Shen et al. [111] studied the effect of PAC-PDMDAAC on the membrane fouling of typical natural organic compounds such as humic acid (HA), bovine serum albumin (BSA) and sodium alginate (SA) by pre-ultrafiltration pretreatment. The decrease in and recovery of membrane flux after backwashing were studied to evaluate membrane fouling. The fouling mechanism of flocs was determined according to particle size and membrane structure. The results showed that the membrane flux increased and the irreversible resistance decreased after PAC-PDMDAAC pretreatment, mainly due to the formation of large size flocs and small fractal dimensions, indicating that PAC-PDMDAAC could effectively alleviate the pollution of the HA, BSA and SA mixture on the membrane. This was a relatively new and expanded study of PDMDAAC in raw water treatment.

#### 5.1.2. Sludge Treatment

Sludge treatment is one of the most important steps in a whole wastewater treatment process. Sludge first needs thorough dehydration treatment before entering the disposal step. At this step, the solid content of sludge is the most critical index. Thus far, the most feasible way to dehydrate sludge has been to use a cationic flocculant. This method facilitated a reduction in the sludge produced and the ease of mechanical dehydration, saving water, improving the effect of sewage treatment and reusing wastewater. It has been reported that P(DAC-AM) and P(DMC-AM) are currently the two most popularly used cationic flocculants in sludge treatment in the world.

As early as 1980, Anon [43] proposed that P(DMC-AM) could be used as a flocculant for sludge treatment in dehydration, which was also the earliest use of P(DMC-AM). Since then, many researchers have applied it to the dewatering treatment of different kinds of sludges. For example, in 2018, Djibrine et al. [50] used their synthesized P(DMC-AM) to treat kaolin wastewater with a turbidity of 117.6 NTU and a turbidity of 5.9 NTU under the best flocculation conditions. The results showed that P(DMC-AM) had a good flocculation performance for the kaolin wastewater, and the charge neutralization and bridging ability could be improved by increasing the cationicity and intrinsic viscosity in a certain range. In 2020, Yang et al. [93] prepared P(DAC-AM) with cationicities of 10~40% and molecular weights of 6~14 million by reversed-phase emulsion polymerization and applied it to the treatment of papermaking wastewater. The experimental results showed that when the cationicity of P(DAC-AM) was 30% and the molecular weight 14 million, the flocculation effect of P(DAC-AM) was the best, with the removal rates of COD and suspended solids attaining levels above 90%.

In the practical application of sludge dewatering, it has been found that on the one hand, the larger the intrinsic viscosity was, the more obvious the flocculation bridging and dehydration effect were; on the other hand, a proper matching of the cationicity of the polymer to the ionicity of the sludge to be treated was critically necessary. Only a polymer with suitable cationicity could destroy the sludge colloid stability effectively, realizing the destabilization and flocculation of colloid particles [112,113]. That is, only the polymer with appropriate cationicity and high intrinsic viscosity could maximize the sludge dewatering effect.

### 5.2. Daily Chemicals

Typically, stable quaternary ammonium salt polymers are the homopolymers and copolymers containing a DMDAAC unit or APTAC (or MAPTAC) unit. Because they have certain sterilization and antistatic abilities due to their quaternary ammonium salt structure and also demonstrate salt and acid resistance due to their unit structure, when used for daily chemicals, these polymers have good stability and can be stably preserved for a long time to be able to fulfill their designated functions.

#### 5.2.1. Polymers Containing a DMDAAC Unit

Due to its stability, low toxicity, and bacteriostatic and antistatic properties, the homopolymer PDMDAAC has been widely used as a regulator, germicidal agent, antistatic agent, and softening agent in skin, hair, nail and tooth care products in daily cosmetics [114,115,116]. For example, as early as 1998, Homola et al. [117] prepared a dental guard containing PDMDAAC, and the guard was applied to cover on a substance. It showed no signs of bacterial infection even after four days of exposure to bacteria. In 2002, Ihikawa et al. [118] used PDMDAAC as one of the main components, combined with a non-ionic surfactant, which could clean clothes well and make clothes bright. In 2017, Masahisa et al. [115] prepared a hair dye conditioner containing PDMDAAC, which could significantly increase the softness of hair moisture retention and toning.

In addition to some homopolymer properties, such as stability and antibacterial and antistatic properties, the copolymer PDA had a much higher and adjustable molecular weight. When used with other components such as salts, surfactants, and care components in daily chemicals (shampoo, soap, hair dye, detergent), it also had thickening properties and improved stability, among other functions. In 2014, Yamazaki [119] formulated a skin cleanser using PDA and an anionic surfactant which could not only wash away dirt but also retain oil on the skin, achieving cosmetic effects. In 2017, Manuel [4] prepared an antistatic shampoo containing PDA which not only did not make the cuticle surface lose its natural emollient but could also improve the smoothness of the hair surface.

#### 5.2.2. Polymers Containing an APTAC (or MAPTAC) Unit

Polymers containing an APTAC (or MAPTAC) unit, similar to the other quaternary ammonium salt polymers, have the advantages of possessing a positive charge and demonstrating a certain compatibility with anionic surfactants, resulting in good performance. Furthermore, because of the acid and alkali resistance and heat resistance of the amide linkage, it can still be stable after a long time under a wide pH range and high temperatures. Thus, it has been increasingly used in skin and hair care products [120,121].

As early as 2000, Wang et al. [122] synthesized a polymer with MAPTAC, acrylic acid (AA), acrylate and AM as raw materials under appropriate conditions. A softener was formulated by using the synthetic polymer and preservatives under pH values of 3.5~4.5, which had better effect when used for hair care compared with the other binary copolymers and terpolymers used in the market. In 2004, Hayakawa [123] prepared a copolymer with vinylpyrrolidone and MAPTAC as raw materials and used it in skincare and hair care cosmetics. This copolymer could remain stable during storage and improve the fluidity, viscosity, moisture retention and emulsion stability of the product. In 2017, Gijsbert et al. [124] synthesized a polymer using APTAC, AA and 2-acrylamide-2-methyl propane sulfonic acid (AMPS) for hair care. It was found that the original smooth feeling remained after multiple shampoos, indicating that it could significantly improve the matting and wet combing of hair and reduce or eliminate the accumulation of chemical substances. In addition, there were many patents reported using the copolymer for shower gels, shampoos, and hair conditioners, as a hair styling agent, in hair oil and as a moisturizer [125,126].

### 5.3. Oil Exploitation

A typical quaternary ammonium salt polymer is a multifunctional chemical agent in the oil field. Polymers containing DMDAAC or APTAC (or MAPTAC) are widely used in drilling, cementing, fracturing, acidification, water injection, water plugging and profile control and tertiary oil recovery due to its acid, alkali, salt, temperature and pressure resistance, especially as an auxiliary agent in the drilling, water injection and tertiary oil recovery fields.

#### 5.3.1. Common Polymers

The structure of DMDAAC is stable, and its polymer had good temperature resistance, salt resistance, acid and alkali resistance. Therefore, it has been widely used as a fluid loss reducer, clay stabilizer for fracturing fluid, and dehydrating agent. DAC (or DMC) polymer was widely used in secondary oil recovery and oilfield wastewater treatment due to its high molecular weight, obvious thickening effect, and flocculation dehydration ability.

In fact, the cationic copolymers used as fluid loss additives for drilling fluids were amphoteric ionic polymers that could be used in various types of water-based drilling fluid systems. Bai et al. [127] developed the novel tetramers of high temperature and filter loss resisting agents with AMPS, AM, DMDAAC and styrene sulfonate (SSS) according to the designed requirements of the molecular structures. At 180 °C for 16 h, the filtration loss of tetramer decreased from 125 mL to 30 mL when the sample dosage increased by 0 from 1.2%, showing that the tetramer fluid loss resisting agent had an excellently high temperature resistance.

In 2015, Lu et al. [128] successfully synthesized a hydrophobic modified cationic flocculant using AM, DMC and methyl methacrylate (MMA) as raw materials in ammonium sulfate aqueous solution by dispersive copolymerization. The flocculant was used in the treatment of oil-bearing wastewater, and its oil-removing performance was evaluated under different conditions. The results showed that the flocculant had obvious advantages in treating oil-bearing wastewater. Similar studies, for example, the study in 2019 by Tian et al. [129], synthesized P(DMC-AM) on the modified composite nanomaterials by reacting activated carbon with DMC and AM and used it to conduct flocculation of an oil-sludge suspension to analyze the flocculation mechanism. The results showed that the material not only had a remarkable flocculation effect, but also realized the flocculation of sludge particles by adsorption bridging and charge neutralization in acidic and alkaline conditions.

#### 5.3.2. Characteristic Polymers

APTAC (or MAPTAC) polymer has become a new type of characteristic polymer used in the oil field because of its advantages of high molecular weight and good acid, alkali, salt, and high-temperature resistance. For example, the United States Exxon Mobil Research and Engineering Company used MAPTAC as a cationic monomer to study multi-function thickeners or gelling agents for acid, alkali, and salt solutions, which were mainly used for acidification and fracturing in oil and gas field development to increase oil and gas production [130].

In 1986, Peiffer et al. [131] prepared an amphoteric ionic polymer as a fluid loss reducer using AM, SSS and MAPTAC as raw materials at 40~60 °C. In the case of 1.0% addition of 3 L water purification mud, the API (30 min) filtration rate was 30 mL, indicating that the prepared amphoteric ionic polymer was a kind of well cement fluid loss reducer with good comprehensive performance. In 2013, Wang et al. [132] synthesized a terpolymer fluid loss reducer for oilfield drilling under high pressure and high temperature using AMPS and MAPTAC as monomers. Under the conditions of 218 °C and 3.44 MPa, when the additive amount of fluid loss reducer was 1.1~1.3%, the fluid loss ratio was 20~30 mL. The results showed that the drilling mud prepared with the fluid loss reducer had good rheological properties and filtrate loss reduction effects after aging at high temperature and high pressure.

### 5.4. Papermaking and Textiles

#### 5.4.1. Papermaking Additive

The use of chemical auxiliaries in papermaking is always an important means to perfect the papermaking process and improve product quality. A variety of polymers, including cationic polymers, play three main roles in the papermaking process: dispersion, enhancement, and flocculation. Therefore, they are used as fiber dispersants, paper dry strengthening agents, wastewater retention and filtration agents, and flocculants.

PDMDAAC was originally used for special conductive paper in the paper industry. Since the 1970s, cationic polymers have been gradually applied in various fields of the paper industry [133]. However, in addition to its use as a flocculant, there were not high requirements on the molecular weight in the process and product application of the paper industry. For example, Shirichiro S [134] reported that adding a film containing PDMDAAC, silicon and other cationic polymers to inkjet printing paper could make the paper white, waterproof, smooth, and high-quality film-forming. Nakamura et al. [135] reported that the addition of PDMDAAC could greatly increase the strength of laminated board. Borkar et al. [136] mixed P(DAC-AM) with polyvinylamine and cassava starch to produce an aqueous coating which could significantly improve the dry strength of paper at low doses. Shan [137] prepared a novel star P(DMC-AM) polymer named S-CPAM by RAFT polymerization. Using the S-CPAM as a paper retentive agent, the effects of different application conditions, such as shear time and pH value, on the retention performance of S-CPAM were investigated. The results showed that S-CPAM significantly increased the retention rate of fine powder and filler through bridging and was insensitive to the shear and pH value of the paper system. All the above studies used low molecular weight cationic polymer.

In 2011, Zhao et al. [138] prepared a series of cationic polymers with high molecular weight by using DMC, AM and N-(2,4,4-trimethylamyl-2-yl) acrylamide as raw materials, and then treated papermaking wastewater as a flocculant in order to measure the chroma and chemical oxygen demand (COD) of treated wastewater samples. The results showed that the polymers could effectively remove chroma and COD in wastewater, and the values were up to 80% and 85%, respectively. In 2020, Yang et al. [139] synthesized a high-concentration solution of P(DAC-AM), whose molecular weight was up to 14~15 million. Then, an emulsion sample was prepared by adding urea, Na_2_SO_4_, and *N*,*N*-dimethyl acrylamide to the polymer solution, which was used for flocculation of the papermaking wastewater treatment. The results showed that after adding the emulsion, the removal rate of chemical oxygen demand (COD) and suspended solids (SS) were 95% and 93%, respectively, with excellent flocculation performance. The above studies on the flocculation treatment of papermaking wastewater used high molecular weight, cationic polymers.

#### 5.4.2. Textile Auxiliary

Cationic polymers have strong hygroscopicity, excellent absorbability and dispersibility, and affinity to fiber. Therefore, they can be used as a color fixing agent, antistatic agent, softener, and fabric finishing agent in the textile printing and dyeing industry. Particularly in recent years, with the wide application of reactive dyes, PDMDAAC as a representative color-fixing agent of reactive dyes without aldehyde was used. The product requires specific molecular weights, but also, in some places, there is more emphasis on molecular weight distribution because the advantages from a specific molecular weight range of the product in its application are more obvious. For example, as mentioned above, Yu et al. [87] measured the color fixing performance of PDMDAAC with different molecular weights on fabric and found that when the molecular weights represented by the intrinsic viscosity were 0.24–0.47 dL/g, its color fixing performance was better.

In 2017, Zhou et al. [140] used a fluorine-containing amphiphilic block copolymer as a stabilizer and DMC as a cationic monomer to prepare a cationic polymer emulsion by the RAFT method. The authors then studied the effect of DMC dosage on fabric finishing properties. It was found that the contact angle between the treated fabric and water decreased with increased DMC dosage, and the treated fabric showed good water repellency.

### 5.5. Other Fields

In recent years, researchers in many fields have been developing new application functions of polymers with cationic quaternary ammonium salt monomers and expanding their application fields. It has been preliminarily reported to be used as a drug carrier in medicine, as a sensor in electronic devices and as an electrode material in fuel cells.

#### 5.5.1. Medicine

Quaternary ammonium salt monomers and their polymers have been used as medical fungicides for more than 100 years [141]. In 2014, Maiti et al. [142] obtained a block polymer by the polymerization of N-isopropyl acrylamide and MAPTAC. It could form a hollow vesicle structure when temperature ≥ 36 °C, wrapping the hydrophobic and hydrophilic macromolecules coumarin-153 and rhodamine 6G, which could be released by lowering the temperature, making it available as a drug delivery vector. Alasino et al. [143] summarized the main properties and biotechnological applications of cationic polymers. This paper showed the numerous effects that could be induced just by a single oral dose of chitosan, either alone or associated with a protein. The researchers thought that cationic polymers should never be considered as an inert, biocompatible component of biotechnological applications aimed to interact with living systems. In 2018, Bistra et al. [144] synthesized reticular PDAC on a 1 mm glass sheet by adding crosslinking agents and used the reticular PDAC as a crosslinking carrier for ibuprofen administration in vivo for a skin safety test. It was found that ibuprofen was released from the drug delivery system in a phosphate buffer solution at 37~38 °C, demonstrating the feasibility of cationic cross-linked polymers as effective carriers for the sustained release of ibuprofen. There have been many similar literature reports [80,81].

#### 5.5.2. Electronic Devices

In 2018, Loehmann et al. [79] synthesized a composite covering polystyrene sodium sulfonate (PSS) and PDMDAAC with a PDMC brush and characterized it at different levels of relative humidity. The relationship between the swelling property and the internal structure of its composites was discussed. It was found that these composites had potential applicability as chemical materials for gas or humidity sensors. Lv et al. [145] designed an electrochemical platform using biocompatible quaternary ammonium salts containing alkyl groups with different chain lengths as electrode materials for visible protein immobilization on a glassy carbon (GC) electrode. This research provided a useful model of quaternary ammonium monomers as promising biomimetic materials with excellent properties and provided a novel platform for the fabrication of biosensors.

#### 5.5.3. Fuel Cells

In 2014, Chen et al. [5] added monomer DMDAAC, a photoinitiator, a crosslinking agent and a microbial solution to a microfluidic device to obtain microbial-coated polymer particles as the anode of a microbial fuel cell. The results showed that the particle-containing microbe could generate voltage for 6 h, which greatly prolongated the substrate loss time. It showed that the cationic-polymer-conducting microparticle had great potential as an anode electrode in microfluidic microbial fuel cells.

## 6. Future Perspectives

In recent decades, with the deepening of research on quaternary ammonium salt monomers and their polymers, the molecular weight of the polymer products has been greatly improved, and the different effects of the structure features on their properties and performances in various application fields has gradually attracted attention. However, there are still many problems to be solved in the optimization of synthetic processes and the research on the relationship between the structure and their properties or performances in industrial applications.

### 6.1. Improvement of the Molecular Weight

The improvement of the molecular weight of polymers is an unchanging topic. High molecular weight means that a wider range of serialized molecular weights of polymer products can be obtained, and products with more fine properties can meet the needs of different application fields, displaying a better application performance. At present, although the molecular weight of most quaternary ammonium salt polymers has been greatly improved, there is still significant room for them to be modified. As an example, acrylamide alkyl quaternary ammonium salts (APTAC and MAPTAC) obviously have space for their weight to be increased when compared with the other two kinds of quaternary ammonium salt polymers (DMDAAC, DAC and DMC). Therefore, researchers can further attempt to improve the molecular weight of APTAC (or MAPTAC) based polymers from the aspects of raw material purity control, selection of polymerization methods and optimization processes to expand their application fields.

### 6.2. The Relationship between the Structure and Properties

The continuous improvement of high and serial molecular weight quaternary ammonium salt polymers provides a good foundation for research into the relationship between its structure and properties. Therefore, the abundant structural information of the polymer molecules, including unit structure, structure composition of chain segments (cationicity), molecular weight and its distribution, can provide many corresponding properties and performance parameters, especially regarding the different physical and chemical properties brought by the polymer’s characteristic functional groups and their molecular weights. In the face of this current situation and development trend, strengthening research on the thermal properties and solution properties of different quaternary ammonium polymers will lay a solid theoretical and experimental foundation for the development of their new functions and application fields.

### 6.3. The Fields of Application of Polymers

Quaternary ammonium salt polymers can be used in traditional fields, such as water treatment, daily chemicals, petroleum exploitation, papermaking, textile printing and dyeing, and in strategic emerging industries, such as pharmaceuticals, photovoltaics, artificial intelligence and new energy materials. The unit structure, proportion of cationic units and the molecular weight of polymers are the internal key factors responsible for their advantages, which can be exploited. It will be important to study the regulations of different polymers suitable for different application fields, analyze and master their working mechanisms, and select the most suitable polymer from the properties of applied objects for practical applications.

## Figures and Tables

**Figure 1 molecules-27-01267-f001:**
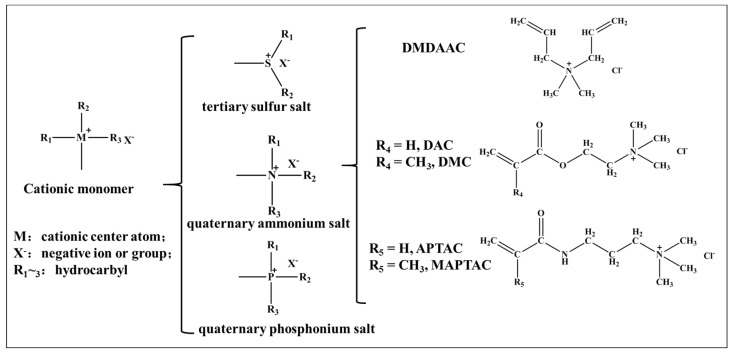
Classification of cationic monomers and structures of representative quaternary ammonium salt monomers.

**Figure 2 molecules-27-01267-f002:**
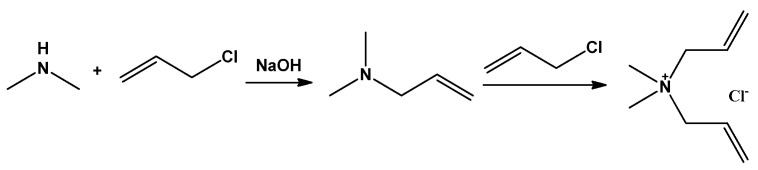
Synthesis routine of DMDAAC.

**Figure 3 molecules-27-01267-f003:**
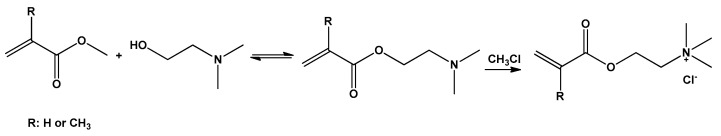
Synthesis routine of DAC (or DMC).

**Figure 4 molecules-27-01267-f004:**
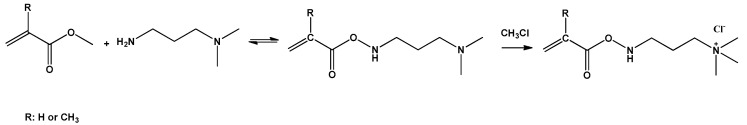
Synthesis routine of APTAC (or MAPTAC).

**Figure 5 molecules-27-01267-f005:**
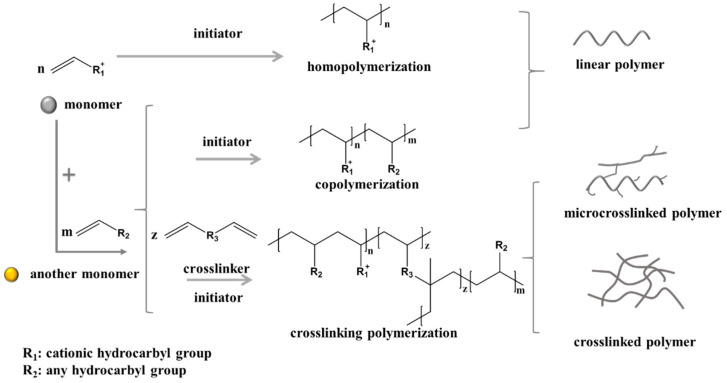
The schematic diagram of cationic monomer polymerization and polymer products.

**Figure 6 molecules-27-01267-f006:**
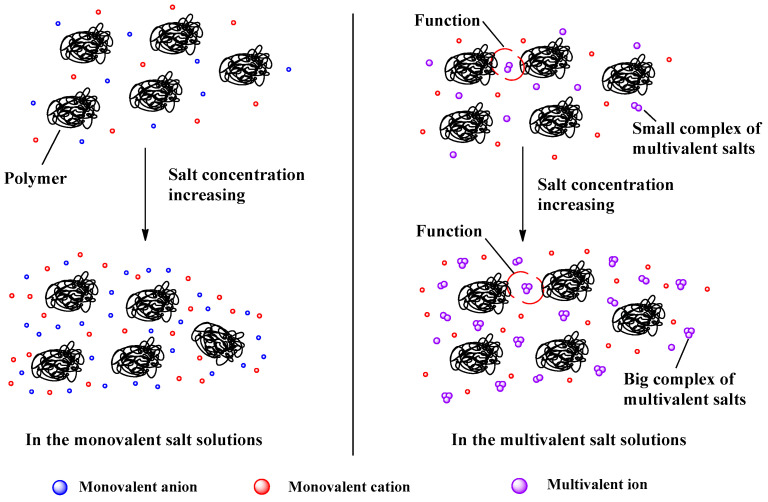
The aggregation model of cationic polymers in salt solutions [91].

**Figure 7 molecules-27-01267-f007:**
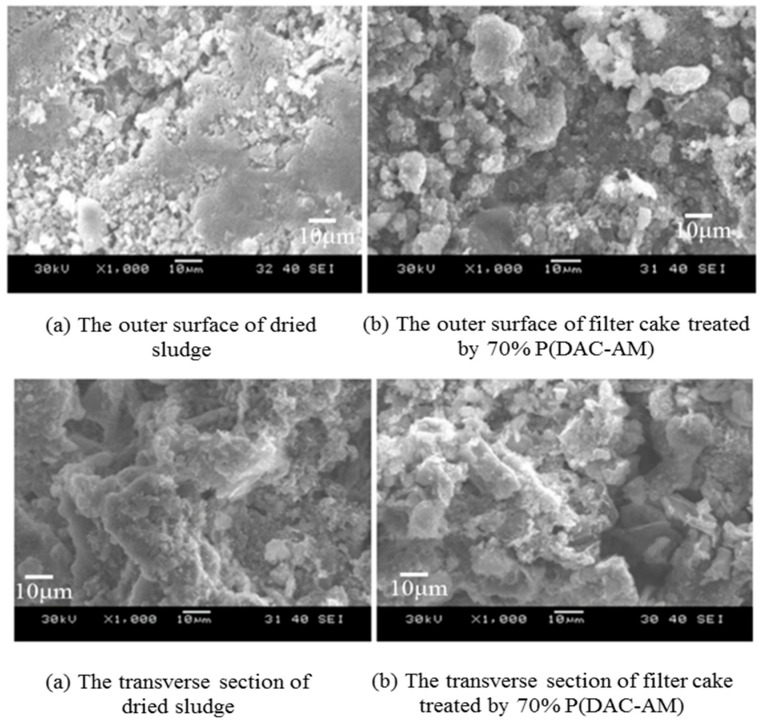
The outer surface and transverse section of dried sludge filter cake treated by P(DAC-AM) [94].

**Figure 8 molecules-27-01267-f008:**
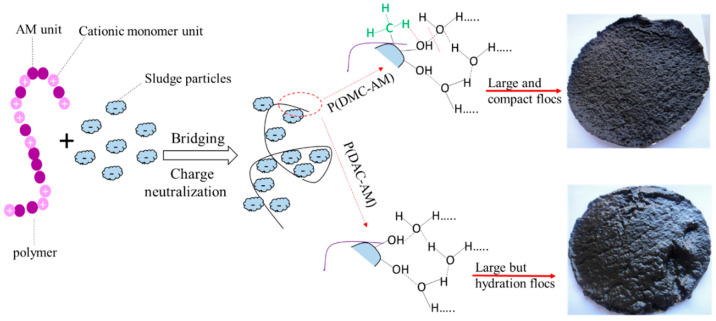
Possible flocculation mechanism of P(DMC-AM) and P(DAC-AM) [92].

**Table 1 molecules-27-01267-t001:** Summarization of the highest intrinsic viscosity quaternary ammonium salt polymers.

Polymer	Researcher	Cationicity/%	Key Conditions	[*η*]/(dL·g^−1^)
PDMDAAC	Ranohoff	-	Co-60, radiation polymerization, 20 h	7.7 (injector)
PDMDAAC	Jia Xu	-	stepwise temperature, 9 h	4.70
PDA	Zhang Y J	5–50	redox initiator, stepwise temperature	21.0–7.0
PDAC	Zhang Y J	-	APS, stepwise temperature	14.6
P(DAC-AM)	Zhang Y J	10–90	redox initiator, stepwise temperature	26.3–15.1
PDMC	Zhang Y J	-	KPS, stepwise temperature	14.5
P(DMC-AM)	Zhang Y J	10–90	compound initiator, stepwise temperature	18.2–14.1
PMAPTAC	Zhang Y J	-	stepwise temperature	8.2
P(MAPTAC-AM)	Li X X	5–50	stepwise temperature	17.2–9.7
PAPTAC	Li X X	-	stepwise temperature	unspecified
P(APTAC-AM)	Li X X	13~75	stepwise temperature	21.5~9.7

**Table 2 molecules-27-01267-t002:** The values of *K* and α in the Mark-Houwink equation [13,95,97].

Polymer	*K*	*α*	Temperature/°C	Condition
PDMDAAC	4.61 × 10^−3^	0.81	25	8.9 × 10^4^ < *M_w_* < 4.7 × 10^5^
PDMDAAC	1.01 × 10^−2^	0.767	30	[*η*] < 1.57 dL/g
PDMDAAC	0.313	0.504	30	1.57 dL/g < [*η*] < 4.49 dL/g
Other typical cationic polymers	4.75 × 10^−3^	0.8	30	high molecular weight polymer
Partially hydrolyzed polyacrylamide	6.98 × 10^−4^	0.91		0.99 dL/g < [*η*] < 2.87 dL/g

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
