# Peer review of "Research Progress on Typical Quaternary Ammonium Salt Polymers"

_molecules, 2022, doi:10.3390/molecules27041267_

Round 1

Reviewer 1 Report

Cationic polymers with quaternary ammonium structures are of great interest for water treatment and drug delivery. In this review paper, the authors first discussed the synthesis of polymers with different chemical structures and macromolecular architectures. They then investigated the structure-property correlation of these polymers and emphasized several key parameters of these polymers. Finally, the discussed various applications of these polymers, such as water treatment, oil exploitation, etc. Overall, this work summarized the recent progress of the synthesis and applications of cationic polymers containing quaternary ammonium salt. However, several aspects of this manuscript needs be to be significantly improved before acceptance.

Comments:

  1. Chemical structures in figure 1-5 need to be significantly improved from the following aspects:
  2. a) For acrylate and methacrylate monomers listed in figure 1, 3 and 4, the authors used H (CH3) to refer to different monomers. However, this is ambiguous and misleading. Instead, the authors should use -R in the structures and list what R groups are below the structure.
  3. b) The bond angles in all the chemical structures need to be fixed.
  4. c) Dimethylaminoethanol structure in figure 3 is not correct.
  5. d) The chemical structures of polymers and network topologies are not clear in figure 5. The authors should clearly point out what polymer architecture would be obtained by using each polymerization method.

  1. The authors should include more figures and schemes in the manuscript. All the figures are related to chemical structures in the current version. The authors should add figures or schemes for section 4 and 5.

  1. The authors need to use the correct technical terms in their discussions. For example, the authors used ‘chain form difference’ in the heading 4.4.2. However, this should be polymer architectures. Also, the authors used’ developing the molecular weight’ in the heading 6.1, which is misused here without a clear meaning. There are more terms in the manuscript besides these two that are not properly used. The authors should check the manuscript and correct their technical terms. Besides, the language in the manuscript needs to be significantly improved with grama errors as well as ambiguous descriptions.

Author Response

Dear reviewer1

Thanks very much for your kind work of my manuscript " Research Progress on Typical Quaternary Ammonium Salt Polymers" (molecules-1540271). We have studied the valuable comments from you carefully and tried our best to revise the manuscript. Revised contents have been marked with red color in the resubmitted manuscript. The replies in terms of questions are listed below.

Q1. Chemical structures in figure 1-5 need to be significantly improved from the following aspects:

  1. a) For acrylate and methacrylate monomers listed in figure 1, 3 and 4, the authors used H (CH3) to refer to different monomers. However, this is ambiguous and misleading. Instead, the authors should use -R in the structures and list what R groups are below the structure.
  2. b) The bond angles in all the chemical structures need to be fixed.
  3. c) Dimethylaminoethanol structure in figure 3 is not correct.
  4. d) The chemical structures of polymers and network topologies are not clear in figure 5. The authors should clearly point out what polymer architecture would be obtained by using each polymerization method.

Answer: Thank you for your valuable comments. We have used –R in the structures instead of H(CH3); The bond angles in all the chemical structures have been fixed; H2O- has been revised to HO- in Dimethylaminoethanol structure in figure 3; We have revised the chemical structures of polymers and network topologies in Fig. 5.

Q2. The authors should include more figures and schemes in the manuscript. All the figures are related to chemical structures in the current version. The authors should add figures or schemes for section 4 and 5.

Answer: Thank you for your suggestion. We added figure 6, 7 and 8 for section 4 at P15 and P18

Q3. The authors need to use the correct technical terms in their discussions. For example, the authors used ‘chain form difference’ in the heading 4.4.2. However, this should be polymer architectures. Also, the authors used’ developing the molecular weight’ in the heading 6.1, which is misused here without a clear meaning. There are more terms in the manuscript besides these two that are not properly used. The authors should check the manuscript and correct their technical terms. Besides, the language in the manuscript needs to be significantly improved with grama errors as well as ambiguous descriptions.

Answer: We have edited the language carefully. ‘chain form difference’ in the heading 4.4.2. has revised to ‘Differences of linear and branched structures; ’ developing the molecular weight’ in the heading 6.1 has revised to ‘Improvement of the molecular weight.

Thank you and best regards.

Yours sincerely,

Yuejun Zhang

2022.1.20

Reviewer 2 Report

  1. It is recommended to concretize the Abstract indicating the key quaternary ammonium salt monomers (or polymers) such as, diallyl dimethyl ammonium chloride (DMDAAC), quaternized dimethylaminoethyl acrylate (DMAEA) (or    dimethylaminoethyl methacrylate, DMAEMA), acrylamidopropyl trimethyl ammonium chloride (APTAC) and methacrylamidopropyl trimethyl ammonium chloride (MAPTAC).
  2. For convenience of readers the review should be provided by Contents and Abbreviation.
  3. All abbreviations should be unified. For instance, such abbreviations as acryloxyalkyl quaternary ammonium salt represented by acryloyl oxygenethyl trimethyl ammonium chloride (DAC) and methylacryloyloxygenethyl trimethyl ammonium chloride (DMC) are the same as DMAEA and DMAEMA.
  4. The review consists of many small Subchapters that distract attention of readers. Also, it contains a lot of repetitive elements to capture the main idea.  In my mind it is better to organize the review concisely in the following manner: 1) Synthetic protocols of cationic monomers (DMDAAC, DMAEA, DMAEMA, APTAC, MAPTAC); 2) Synthesis of quaternary ammonium salt polymers based on DMDAAC, DMAEA, DMAEMA, APTAC, MAPTAC. In this case it is not necessary to describe separately the Chapter 3 devoted to research progress on synthesis of quaternary ammonium polymers. This part can be dissolved in paragraph 2. Chapter 4 describes the relationship between structure and properties of cationic polymers. However, I do not see any information on solution properties of described polymers in aqueous and aqueous-salt solutions. The hydrodynamic, conformational and molecular characteristics of polymers are absent. At least one Table summarizing the molecular weight, intrinsic viscosity, polydispersity index, constants of Mark-Kuhn-Houwink equation should be given.
  5. Chapter 5 devoted to application of quaternary ammonium polymers should be concisely rewritten. I suggest to concentrate the attention of authors to Water purification, Oil recovery, Paper and textile industry. Application aspects of quaternary ammonium-based polymers in biotechnology and medicine were comprehensively described in review articles (Y. Jiao et al. Progress in Polymer Science, 71 (2017), 53–90. http://dx.doi.org/10.1016/j.progpolymsci.2017.03.001; Construction of a quaternary ammonium salt platform with different alkyl groups for antibacterial and biosensor applications. RSC Adv., 2018, 8, 2941, DOI: 10.1039/c7ra11001d; Roxana V. Alasino, Karina L. Bierbrauer, Dante M. Beltramo, Silvia G. Correa and Ismael D. Bianco. Cationic Polymers for Biotechnological Applications. Frontiers in Biomaterials, Vol. 2, 2016, 3-27). Unfortunately, these Refs are not cited by authors.
  6. The text is difficult to read. The manuscript is written in poor English. It contains many grammatical and stylistic errors. Please attract native English speaker for editing.
  7. Lines 36, 37. The sentence “According to the ionic properties of hydrophilic groups, water-soluble polymers can be divided into cationic, anionic, and non-ionic types” is wrong because “non-ionic types” should be replaced by “amphoteric types”.
  8. Lines 145-147. Probably “mothed” means “method”.
  9. Line 149. Should be corrected as “It means…”

Author Response

Dear reviewer2

Thanks very much for your kind work of my manuscript " Research Progress on Typical Quaternary Ammonium Salt Polymers" (molecules-1540271). We have studied the valuable comments from you carefully and tried our best to revise the manuscript. Revised contents have been marked with red color in the resubmitted manuscript. The replies in terms of questions are listed below.

Q1. It is recommended to concretize the Abstract indicating the key quaternary ammonium salt monomers (or polymers) such as, diallyl dimethyl ammonium chloride (DMDAAC), quaternized dimethylaminoethyl acrylate (DMAEA) (or    dimethylaminoethyl methacrylate, DMAEMA), acrylamidopropyl trimethyl ammonium chloride (APTAC) and methacrylamidopropyl trimethyl ammonium chloride (MAPTAC).

Answer: Thank you for your valuable comments. The key words ”diallyl dimethyl ammonium chloride; acryloyl oxygen ethyl trimethyl ammonium chloride; acrylamide propyl trimethyl ammonium chloride;” have been added.

Q2. For convenience of readers the review should be provided by Contents and Abbreviation.

Answer: Thank you for your suggestion. Contents and Abbreviation have been provided in manuscript.

Q3. All abbreviations should be unified. For instance, such abbreviations as acryloxyalkyl quaternary ammonium salt represented by acryloyl oxygenethyl trimethyl ammonium chloride (DAC) and methylacryloyloxygenethyl trimethyl ammonium chloride (DMC) are the same as DMAEA and DMAEMA.

Answer: Thank you for your suggestion. All abbreviations have been unified.

Q4. The review consists of many small Subchapters that distract attention of readers. Also, it contains a lot of repetitive elements to capture the main idea.  In my mind it is better to organize the review concisely in the following manner: 1) Synthetic protocols of cationic monomers (DMDAAC, DMAEA, DMAEMA, APTAC, MAPTAC); 2) Synthesis of quaternary ammonium salt polymers based on DMDAAC, DMAEA, DMAEMA, APTAC, MAPTAC. In this case it is not necessary to describe separately the Chapter 3 devoted to research progress on synthesis of quaternary ammonium polymers. This part can be dissolved in paragraph 2. Chapter 4 describes the relationship between structure and properties of cationic polymers. However, I do not see any information on solution properties of described polymers in aqueous and aqueous-salt solutions. The hydrodynamic, conformational and molecular characteristics of polymers are absent. At least one Table summarizing the molecular weight, intrinsic viscosity, polydispersity index, constants of Mark-Kuhn-Houwink equation should be given.

Answer: Thank you for your suggestion. On one hand, because the synthesis route, method, purpose and evaluation index, etc. for monomer and polymer are completely different and we have felt very difficult to put them in one room, so we don’t merge paragraph 2 and 3 together, even though we have considered for a quite long time, already. On the other hand, we add the part “4.3. Solution property of polymer”, which include the information on solution properties of described polymers in aqueous and aqueous-salt solutions.

Q5. Chapter 5 devoted to application of quaternary ammonium polymers should be concisely rewritten. I suggest to concentrate the attention of authors to Water purification, Oil recovery, Paper and textile industry. Application aspects of quaternary ammonium-based polymers in biotechnology and medicine were comprehensively described in review articles (Y. Jiao et al. Progress in Polymer Science, 71 (2017), 53–90. http://dx.doi.org/10.1016/j.progpolymsci.2017.03.001; Construction of a quaternary ammonium salt platform with different alkyl groups for antibacterial and biosensor applications. RSC Adv., 2018, 8, 2941, DOI: 10.1039/c7ra11001d; Roxana V. Alasino, Karina L. Bierbrauer, Dante M. Beltramo, Silvia G. Correa and Ismael D. Bianco. Cationic Polymers for Biotechnological Applications. Frontiers in Biomaterials, Vol. 2, 2016, 3-27). Unfortunately, these Refs are not cited by authors.

Answer: Thank you for your valuable comments. We concentrate to Water treatment, Oil exploitation, Papermaking and textile industry, and refers to other fields including medicine, electronic device and fuel cell. The references referred by reviewer have been cited.

Q6. The text is difficult to read. The manuscript is written in poor English. It contains many grammatical and stylistic errors. Please attract native English speaker for editing.

Answer: We have revised the grammar errors and modified the stylistic problems once again, carefully..

Q7. Lines 36, 37. The sentence “According to the ionic properties of hydrophilic groups, water-soluble polymers can be divided into cationic, anionic, and non-ionic types” is wrong because “non-ionic types” should be replaced by “amphoteric types”.

Answer: Thank you for your valuable comments. “Amphoteric” has been added.

Q8. Lines 145-147. Probably “mothed” means “method”.

Answer: Thank you for your valuable comments. “ method” has been revised.

Q9.Line 149. Should be corrected as “It means…”

Answer: Thank you for your valuable comments. Line 149. has been revised to “It means…”

Thank you and best regards.

Yours sincerely,

Yuejun Zhang

2022.1.20

Round 2

Reviewer 1 Report

The authors have addressed some of my comments in this manuscript, however some questions remain after revision.

  1. Small molecules and polymers have typical bond angles, which should be represented by the chemical structures in the figures. However, most of structures in figure 1-5 still showed incorrect angles. The authors should correct it automatically using the software. Answer: Thank you for your valuable comments. We have used ChemBioDraw to draw the structures, and fixed the bond angles. Then, used “clean up structure” order to correct the structure automatically.

2. The languague still needs to be significantly improved. This has been pointed out by both reviewers. Although languaguage is improved in the manuscript, there are still many errors in the manuscript and many contents that are hard to understand. The authors should ask a native speaker to carefully polish the language.

Thanks very much for your kind work of my manuscript " Research Progress on Typical Quaternary Ammonium Salt Polymers" (molecules-1540271). We have studied the valuable comments from you carefully, and tried our best to revise the manuscript.